# How do emergency department staff respond to behaviour that challenges displayed by people living with dementia? A mixed-methods study

Laura Goodwin ,[1] Cathy Liddiard,[1] Sera Manning,[1] Jonathan Richard Benger,[1] Edward Carlton,[2] Richard Cheston,[1] Rebecca Hoskins,[3] Hazel Taylor,[4] Sarah Voss [1]

[1]School of Health and Social Wellbeing, University of the West of England, Bristol, UK
[2]Emergency Department, North Bristol NHS Trust, Westbury on Trym, UK
[3]Emergency Department, University Hospitals Bristol and Weston NHS Foundation Trust, Bristol, UK
[4]Research Design Service, University Hospitals Bristol NHS Foundation Trust, Bristol, UK

**Correspondence to**
Dr Laura Goodwin;
Laura.goodwin@uwe.ac.uk

## ABSTRACT

**Objectives** To investigate staff experiences of, and approaches to behaviour that challenges displayed by patients with dementia in the emergency department (ED). Behaviour that challenges is defined as 'actions that detract from the well-being of individuals due to the physical or psychological distress they cause within the settings they are performed', and can take many forms including aggressive physical actions, shouting and verbal aggression and non-aggressive behaviour including repetitive questioning, inappropriate exposure and resistance to care.

**Design** Mixed-methods study consisting of an online survey and semistructured telephone interviews. Quantitative data were analysed and presented using descriptive statistics. Qualitative data were analysed thematically.

**Setting** The EDs of three National Health Service (NHS) Hospital Trusts in Southwest England.

**Participants** Multidisciplinary NHS staff working in the ED.

**Results** Fifty-two online survey responses and 13 telephone interviews were analysed. Most (24/36, 67%) survey respondents reported that they had received general training in relation to dementia in the last 2–5 years, however, less than a fifth (4/23, 17%) had received any ED-specific dementia training. All (48/48) felt that behaviour that challenges could potentially be prevented, though resource constraints and practice variation were identified. Four main themes emerged from the qualitative data: (1) the 'perfect storm' of the ED; (2) behaviour that challenges is preventable with the right resources; (3) improvisation and (4) requirement for approaches that are specific to the ED.

**Conclusions** The findings of this study suggest that ED staff do not feel that they are prepared to respond effectively to behaviour that challenges displayed by people living with dementia. Future work could adapt or develop an intervention to support ED staff in responding to behaviour that challenges.

## BACKGROUND

In the UK, there are around 944 000 people living with dementia, and this is estimated to increase to more than 1.6 million by 2050.[1] Challenging behaviour, also referred to by the Alzheimer Society and others as 'behaviour that challenges' (BtC),[2] is commonly associated with dementia and has been defined as 'actions that detract from the well-being of individuals due to the physical or psychological distress they cause within the settings they are performed'.[3] BtC can take many forms including aggressive physical actions (eg, kicking, biting), shouting and verbal aggression and non-aggressive behaviour including repetitive questioning, inappropriate exposure and resistance to care. This behaviour often stems from an unmet need—for instance, to reduce distress or frustration or to maintain a sense of well-being. Both the person living with dementia and the people around them are likely to be impacted by their behaviour.[4]

An increase in the number of people living with dementia is reflected in emergency hospital admissions. In England, there was a rise of 35% in emergency admissions for people with dementia during 2017–2018, compared with 2012–2013.[5] More than half of the people who had a diagnosis of dementia

in 2017–2018 were admitted to hospital as an emergency at least once in that year.[5]

In a busy emergency department (ED), the physical and medical aspects of care such as completing observations and administering medication may become prioritised over less technical elements (eg, helping with toileting or food/drink),[6] and this may result in unmet needs for older people living with dementia.[7] For instance, the loud and often crowded environment of the ED can result in sensory overload for people living with dementia,[8 9] and may lead to dehydration and ineffective pain management, exacerbating other unmet needs with an increased risk of BtC.[10–12]

BtC can be difficult to respond to for ED staff, who need to ensure safety and prompt treatment for all patients. In a retrospective audit by Tropea et al,[13] 39% of people living with dementia attending an ED in Australia exhibited BtC, with aggression being the most common (35%). Despite the importance of dementia training being recognised by Royal College of Emergency Medicine,[14] there are no standardised processes for managing patients living with dementia or cognitive impairment in the UK EDs. Without training and resources to prevent or respond to challenging behaviour associated with dementia, ED staff have previously relied on sedating or restraining patients, which is often distressing for patients, family members and staff and likely to lead to adverse outcomes.[13 15]

Various approaches are available to hospital staff for responding to BtC for people living with dementia: creating a calm, quiet physical environment[16]; sensory stimulation such as light and music therapy[17]; distraction activities such as video, music boxes or dolls[18]; accessing 'hospital ready' information about the patient and their wishes.[19 20] However, there is little research on the effectiveness of prevention or de-escalation interventions in EDs,[15] where there may be unique challenges to implementing some of these approaches. This means that there is no standardised use of these approaches across EDs in the UK. Therefore, this study aimed to investigate UK National Health Service (NHS) staff experiences of responding to BtC, associated with dementia, in the ED. The aims of this project were:

1. To explore staff experiences of and approaches to BtC associated with dementia in the ED environment.
2. To explore ED staff opinions of using de-escalation techniques within an ED setting and any perceived barriers and facilitators to introducing new interventions.

## METHODS
### Setting
The setting was the EDs in three NHS Hospital Trusts in the Southwest of England: University Hospitals Bristol and Weston NHS Foundation Trust (UHBW), North Bristol NHS Foundation Trust (NBT) and Royal United Hospitals Bath NHS Foundation Trust (RUH). ED attendance at each site is as follows: over 90 000 people a year at NBT; over 100 000 people a year at UHBW; over 70 000 a

year at RUH.[21] There are estimated to be more than 2574 people in Bath and North East Somerset currently living with dementia, and 4500 people living with dementia in Bristol.[22 23] All participating Trusts currently use the 'This is Me'[20] document for attending patients who are known to be living with dementia.

### Patient and public involvement
Patient and public involvement (PPI) was gained from a local support group for people living with dementia and their carers in Bristol, who have experience of attending local EDs as/with a person living with dementia. The group provided feedback on the study aims, design and recruitment, and supported the research team in designing all patient-facing documents.

### Study design
Mixed-methods study consisting of an online survey and semistructured interviews. Both the survey and interviews were open to all staff working in the three NHS Hospital Trust EDs above. Convenience sampling was used: an email promoting the study (and including the survey link) was sent to all ED staff by local collaborators at each Trust and disseminated through bulletins and posters with a quick response (QR) code for the survey displayed in ED staff rooms at participating sites. Recruitment adverts included an offer of a £10 voucher for those who completed an interview.

The survey was hosted on the 'Qualtrics' platform (Provo, Utah) and accessed through the disseminated links and QR codes. Participants were required to confirm consent electronically before the first survey question was presented. As the survey was exploratory, a sample size calculation was not required.

Eligible participants were invited to contact the lead author to take part in an interview. Potential interview participants received a participant information sheet, privacy notice and consent form via email. Verbal confirmation of consent was audiorecorded at the start of the telephone interview. We aimed to recruit 10–15 participants, guided by information power with a narrow aim and specific study population.[24]

### Data collection
#### Survey
The open online survey (online supplemental file 2) was developed by the study team from the findings of a literature review on responding to BtC in people with dementia in the ED,[25] in collaboration with ED colleagues. It was then reviewed by members of our PPI group, and piloted for usability and technical functionality by members of the study team. The survey was open between November 2020 and November 2021. A participant information sheet, privacy notice and consent form were integrated into the first page of the survey. The survey contained questions about the extent and nature of the issues associated with caring for patients with dementia whose behaviour may be challenging in an ED setting. Questions with open

free-text responses were included alongside questions with Likert scales and multiple-choice responses.

## Interviews

The participant information sheet and consent form were developed in collaboration with the study team and our PPI group. The interview topic guide (online supplemental file 1) was drafted by the study team, based on existing literature and the study team's expertise. It then was reviewed by our PPI group, and piloted with clinical colleagues. Semistructured interviews were used to investigate staff experiences of responding to BtC associated with dementia in the ED and any prevention/de-escalation techniques used. A single interview was held with each participant, which was conducted via telephone at a time convenient for the participant. Interviews were held between October 2020 and November 2021 by experienced qualitative researchers (LG and CL) and audio-recorded via Skype for Business. Interviews lasted an average of 45 min. Audio files were transcribed verbatim.

## Analysis

Survey data were exported from Qualtrics to a Microsoft Excel spreadsheet. Quantitative data were analysed and presented using descriptive statistics. Free-text data were summarised and categorised.

Interview transcripts were coded and analysed by CL and LG, supported by NVivo V.12 software. Data were analysed thematically through a well-established iterative process of data reduction, constant comparison, organisation and understanding.[26] The researchers read the transcripts several times and then coded selections of text to represent instances of a concept. Codes were then reviewed in terms of their relationship to other codes and combined to create more developed themes. From this analysis, distinctions could be made between the different levels of themes.

## RESULTS
### Participant characteristics

A total of 52 responses to the survey were received from ED staff (table 1). Thirteen interviews were conducted (table 2). The largest proportion of survey participants (22/52, 42%) were doctors, while the majority of telephone interviewees (8/13, 62%) were nurses. We did not collect data on gender for the survey, but the majority of telephone interviewees (10/13, 77%) were female.

Four main themes were identified from the qualitative data: (1) the 'perfect storm' of the ED; (2) BtC is preventable with the right resources; (3) Improvisation and (4) requirement for approaches that are specific to the ED. These are reported in narrative form, supported by quotes and woven into the quantitative results below.

### Theme 1: the 'perfect storm' of the ED

The significant challenges that the ED setting presents to people with dementia was described by all interview

**Table 1** Survey respondent characteristics (n=52)

| Characteristic | No (%) of respondents |
|---|---|
| **Role** | |
| Doctor | 22 (42) |
| Nurse | 19 (37) |
| Admin staff | 1 (2) |
| Porter | 1 (2) |
| Other | 9 (17) |
| **Approximate experience working in ED (years)** | |
| <1 | 6 (12) |
| 1–2 | 13 (25) |
| 3–5 | 8 (15) |
| 6–10 | 9 (17) |
| 11–15 | 8 (15) |
| 16–20 | 5 (10) |
| 21+ | 3 (6) |

ED, emergency department.

participants. Survey data indicated that almost all (49/52, 94%) respondents had experienced BtC in the ED from patients living with dementia. Telephone interviewees perceived the 'perfect storm' of the ED to be made up of triggers (physiological and environmental), the ED's clinical approach to patients and a lack of resources.

The patient being in pain was most commonly ranked as the number one trigger for BtC by survey participants (14/46, 30%). During telephone interviews, however, most participants described the hospital environment itself as the dominant triggering factor for BtC due to an absence of familiar faces and routine. Most participants reported that there to were factors unique to the ED (in comparison to other hospital settings) which compounded patients' unmet needs such as: constant

**Table 2** Interview participant characteristics (n=13)

| Characteristic | No (%) of respondents |
|---|---|
| **Role** | |
| Doctor | 4 (31) |
| Nurse | 8 (62) |
| Other | 1 (7) |
| **Gender** | |
| Female | 10 (77) |
| Male | 3 (23) |
| **Approximate experience working in ED (years)** | |
| ≤3 | 4 (31) |
| 4–6 | 5 (39) |
| 7–9 | 2 (15) |
| 10+ | 2 (15) |

ED, emergency department.

bright lighting; the busy nature of the department; long waits; open areas that pose a risk of absconding; staff having less time to get to know their patient.

> It's noisy, it's bustling, it's confusing, you [the patient] often feel isolated, there are people walking past you all the time looking at you. There are strangers coming in and out. I mean everything in ED plays against dementia. You know, for those of us that are used to the chaos of the place it can still be challenging on a bad day, but to somebody with dementia it must be a nightmare. It's got everything going against it. (P011)

Inadequate information transfer was also highlighted as a resource issue for effectively managing BtC in the ED. Interview participants reported incomplete handovers from ambulance staff, absent personalised care plans and less chance of family/friends being able to communicate the patient's needs, due to restrictions on visitors since related to the COVID-19 pandemic.

Some participants described how the primary function of the ED is to focus on the patient's presenting clinical condition (eg, stemming the bleeding from a fall or providing treatment for blood pressure), which meant that they were often unable to remove the environmental triggers in the ED for patients living with dementia, as they were necessary for providing safe medical care.

> Everybody needs to back off and leave them for a while because I think any form of intervention only escalates the situation. The trouble is, of course, we are under time pressures and our focus is on getting the injury sorted. (P011)

Within this 'perfect storm', telephone interviewees recalled numerous experiences of BtC in the ED, with many reporting instances occurring on every shift. Survey data reflected the telephone interviews, where the majority of participants (36/46, 78%) reported that BtC occurred on a weekly basis in their ED. The most common types of behaviour reported by survey participants included 'confusion' (45/46; 98%), 'wandering' (43/46; 94%), 'agitation' (43/46; 94%) and 'verbal aggression' (42/46; 91%). Telephone interview participants described instances of all of these behaviours within their ED, and reported concerns regarding patient safety and staff resource.

> The most frequent experience that happens with a patient with dementia is that they come into the emergency department and they wander, and unfortunately we don't have a dedicated member of staff to stay with them. So, it's scary because as a nurse you feel that they are in danger. (P007)

Some behaviours were reported as being more challenging in an ED environment, and participants expressed anxiety around gaps in dementia care in the ED.

> I don't think there's any areas that are 100% safe. I know you can't be with dementia but it's quite risky

in ED. But then they're only supposed to be there for four hours and then go onto a ward. But that obviously doesn't always happen with, you know, lack of beds. (P006)

Staff interviewed via telephone reported that prevention and de-escalation of BtC could be difficult to maintain in the ED environment and that, if responded to ineffectively then behaviours found to be challenging by staff could continue throughout the patient's entire ED stay.

> Nine times out of ten [the BtC lasts] until they get to a ward, or until they are picked up by transport. (P001)

> I would say there are nights where we don't really get people settled at any point. (P012)

Some of these interviewees described how the challenge of managing BtC worsened at night, as people living with dementia often experienced sleep deprivation and were unable to settle in the ED environment. One reported the rate of falls to be higher at night, and noted difficulties escalating patients or getting support from dementia teams during this time.

### Theme 2: BtC is preventable with the right resources

Survey respondents indicated they felt that instances of BtC could either be prevented (13/48 30%) or sometimes prevented (35/48, 73%) in the ED. One-to-one care from staff or a carer was ranked as the intervention most likely to prevent BtC (18/44, 41%), followed by rapid assessment of the patient (10/44, 23%). For de-escalating BtC, one-to-one care was also the highest-ranking technique (25/44, 57%), followed by the provision of a distraction (7/44, 16%) or a quieter environment (7/44, 16%). However, participants in both the survey and interviews highlighted the lack of resources available to provide these types of interventions, noting concerns regarding crowded EDs, understaffing and long ambulance wait times.

> It's the endless resources isn't it, that we don't have? If there were a team of Dementia Nurses in the hospital that you know, Healthcare Assistants even, that could just come and sit with the patients, engage them, in looking at that magazine, listening to that music, you know, or whatever it is, if it's just sitting and chatting with them. (P009)

Despite a common acknowledgement of the unique challenges of the ED for people living with dementia, less than a fifth (4/23, 17%) of survey respondents had received dementia training that was specific to the ED context. None of the telephone interviewees reported having had any ED-specific dementia training.

While most survey respondents (24/36, 67%) reported that they had received general training in relation to dementia in the last 2–5 years, a large proportion (11/23, 48%) found it only 'moderately useful'. This was reflected

**Table 3** Existing dementia-friendly initiatives recalled by interview participants

| Initiative | Limitation(s) |
|---|---|
| Reminders of the past: designing corridors with pictures of past-times and/or old films played on a TV | ► They become out of date. |
| Colouring books | ► Not always in stock. |
| Box of distractions | ► Requires space/tables to use. <br> ► Can require a staff member to facilitate their use. <br> ► Not available during COVID-19 as has to go through infection control. |
| Providing food and drink | ► Short-term solution <br> ► Hot food not always available in ED. <br> ► White plates can cause confusion with some (white) foods. |
| 'This is Me'[20] or similar: document containing information about the patient's needs and wishes | ► Patient does not always come in with it. <br> ► Specific paperwork required to create one is not in stock. <br> ► No time to create one in ED setting. <br> ► Suggestions within it may not be appropriate for the ED setting. |
| Music | ► Equipment went missing. |
| Family member/carer | ► COVID-19 restrictions reduced this. |
| Individual cubicles | ► Not always safe, as unable to see the patient. <br> ► Many cubicles not available as used for patients with COVID-19 or other infectious conditions |

ED, emergency department.

in interview data, where general dementia training was given mixed reviews in terms of its usefulness.

Most telephone interviewees could recall generic dementia-friendly initiatives that had been implemented in their ED. One of the initiatives, music, was enthusiastically highlighted as being particularly helpful to patients, although frustration was expressed at how quickly equipment would go missing. Other de-escalation techniques were also perceived as useful to staff and patients, but telephone interview participants noted that such techniques often needed an investment of time and staff to facilitate them, which was not always possible. Indeed, limitations (most often linked to resource issues) were described for most of the initiatives described positively by staff (table 3).

The majority of survey respondents (30/45, 67%) felt that there should be a separate assessment area of the ED for patients with dementia. Free-text comments added that the ideal space should be quiet, where the lights can be dimmed and attended by staff trained in dementia. From respondents who did not think there should be a separate assessment area (33% 15/45), free-text comments showed support for this type of area in an 'ideal world' but reported that the logistical and resource challenges this would present to enable patients to be safe there was likely to render the initiative unfeasible.

### Theme 3: improvisation
While most survey respondents (23/36, 64%) reported that they covered information about how to manage BtC in their general dementia training, there was variation in responses to BtC by the ED staff surveyed. The most used

approaches to resolving or reducing the impact of BtC were distraction techniques (37/46, 81% reported using them 'always' or 'most of the time'), and the use of family and/or carers (20/46, 43% reported using them 'always' or 'most of the time').

Some telephone interviewees reported using their own individual approaches to preventing and de-escalating BtC, learnt from other professional roles or their personal life. One described 'messing-up' sheets and asking patients to fold them, or offering bandages to squeeze while carrying out a procedure to keep their hands occupied.

> Everyone uses their own sort of techniques. No-one's really said you can or can't do this, but you just have to improvise with what you've got and it's difficult when you haven't got a lot to help you. (P011)

Telephone interview participants reported that an inconsistent approach to managing BtC often resulted in ineffective or non-sustained management of behaviour within the ED, and that this became particularly problematic when it led to patients resisting medical care. Indeed, the majority of survey respondents (87%, 40/46) reported that they had experienced physical resistance to care when supporting people living with dementia in the ED. Participants interviews via telephone reported that this often meant that procedures were either minimised or postponed, or described some kind of 'battle' with the patient to complete the procedure; causing distress for both patient and staff.

I had a lady yesterday, who…needed a Lateral Flow test. So, you know, to go and stick a thing up her nose, she didn't want that done. And it was a horrible thing to have to do and, in the end, I gave up with it, because it was just distressing her too much… But yeah, she was grabbing and pinching [me] and you know, she, she was quite up for giving you a hearty thump at the end of it. (P009)

When asked directly about the use of physical restraint as a response to BtC in the ED, most survey respondents (36/46, 78%) reported that they never used it. However, there seemed to be some differences in how the word 'restraint' was interpreted by telephone interview participants, which often led to contradictory accounts during interviews. While most telephone interviewees said that restraint was only very rarely used, when questioned directly, many described using or witnessing some form of restraint frequently (to administer care) when talking more generally about their experiences of managing BtC in the ED.

I've not seen anyone have to be physically restrained or anything like that. I think we try to kind of implement things in place before anything like that would need to happen. (P002)

For simple things like taking a blood test…you'll often hold the patient's hand down, or arm down, and they can't consent. (P005)

There was also a contradiction between survey and interview responses when reporting the use of pharmaceutical measures to manage BtC in the ED. Telephone interviewees were unified in their perception that the use of pharmaceutical measures to reduce BtC in the ED was very rare, while the majority (38/46, 83%) of survey respondents reported that this approach was 'sometimes' used. Again, there seemed to be some differences in the interpretation of the wording, which meant that mild sedation to complete a medical procedure was not always viewed as a pharmaceutical measure to manage BtC.

Some telephone interview participants described an accelerated approach to completing medical assessments/procedures to minimise the chance of the patient showing physical resistance to care, even if this meant completing tasks without explicit consent.

It's horrendous, but you just do things. You almost don't ask, if you're doing these things… I might say, 'oh I'm going to exam your eyes, that's okay?' But you just roll off the natural things, and the nurses do them, and you're doing it out of best interest, but sometimes it's easier to be like 'oh, they don't necessarily understand' and you just crack on and do it… You're just trying to be, like, she probably won't understand the blood test and things, but I think she really needs it, she might have a raging infection, we need to do this. (P005)

### Theme 4: requirement for approaches that are specific to the ED

There was overwhelming support from telephone interview participants regarding the potential introduction of an ED-specific intervention to support staff in managing BtC. In fact, one participant had developed their own dementia training package, specific to the ED. This training aimed to encourage staff to consider the potential trigger(s) of the BtC (ie, toileting needs, discomfort/pain) and ways to minimise these triggers, while the patient was in the ED.

A number of participants put forward the idea of a simple check-list or escalation toolkit for care, which could use a 'stepwise approach that will guide you in both recognising and dealing with these behaviours' (P004). Participants said that a formal tool would increase the consistency of approach to BtC across the ED and provide something to follow when behaviour was escalating. One interviewee described their vision of what an 'escalation tool' could look like:

If it was like a scale of one to ten, the first thing is make sure they're not in pain and then the second thing is do they need the toilet, almost like a tick box of things you need to consider, and then you work your way up…from basic things to, kind of, more difficult things to implement. Then you could tick them off almost—what you've done and at what point it de-escalated. So then it would potentially help with their care in the future when they're on the wards or if they came back into ED. (P002)

Another suggested that a new intervention should give more emphasis to preventing triggers for BtC, rather than focusing solely on de-escalation.

I guess the bit we don't talk about is preventing those behaviours happening in the first place, and how to make our department a place that prevents things escalating…We wait for it to escalate and then we think about, is this patient comfortable? Are they in a quiet enough environment? Do they have food and drink? (P012)

Other suggestions included better identification of patients who may be living with dementia, specialist ED-specific training for staff, and improved co-ordination with other services and organisations. For example, participants reported that there could be better handover from care-homes and ambulance services, including multiservice utilisation of an electronic version of the 'This is Me'[20] document, or arranging for home-comforts to be brought to the ED with the patient.

We don't think enough about them having something from their home or from their care home, like a blanket or something that they find familiar, that has the right smell. (P007)

There were some potential barriers put forward by ED staff when implementing a new intervention. Telephone interview participants advised that any new initiative

should not need too much space or time, be complex, or *'another piece of paperwork to fill out'* (P012). They also warned that measuring the impact of any new intervention may be challenging.

Facilitators to a new intervention, put forward by telephone interview participants, included buy-in from key stakeholders and emphasis on how the intervention would help patients. One interviewee suggested the use of face-to-face study days to explain the need for a new tool, while another emphasised the need for any information to be distributed to all ED staff who would come into contact with patients (ie, including porters, receptionists, security guards and healthcare assistants).

## DISCUSSION

To the best of our knowledge, this is the first study to specifically focus on staff experiences of responding to BtC in the ED environment in the UK. A mixed-methods study design combining survey and qualitative interviews provided both a wide overview as well as a rich description of staff experiences. Important weaknesses of our study include the relatively low response rate and the fact that the research was confined to one geographical area which will have limited the representativeness of our findings. We used a convenience sample of volunteers, and so the experiences and views on which our research is founded may not be representative of all ED staff.

We found several challenges associated with responding to BtC associated with dementia in the ED. These included the concept of the ED as a 'perfect storm' for behavioural triggers, a lack of resources to effectively prevent and de-escalate BtC within the ED, and inconsistent approaches to managing BtC, leading to escalation of behaviour. Staff were positive about the idea of introducing ED-specific interventions to prevent/de-escalate BtC and offered a number of suggestions for what this might include.

The description of the ED as a 'perfect storm' is well supported in the literature regarding dementia.[27] Our findings suggest a lack of ED-specific training and resources. While a number of different interventions have been found to be helpful in the general hospital setting, the effectiveness of these approaches in the ED is unknown due to a lack of research.[15] Participants in our study noted that a lack of ED-specific training and resources led to variation in the techniques used by staff to respond to and de-escalate BtC, which in turn led to ineffective management; staff were left to improvise with the resources they had, often attempting to intervene once behaviours had already escalated substantially.

In line with previous literature, our findings suggest that ineffective management of BtC sometimes resulted in physical and pharmacological restraint being used by staff in the ED.[13] However, there seemed to be differences in the interpretation of words such as 'restraint' and 'sedation' by participants; some staff did not perceive physical measures used to facilitate minor procedures as

formal restraint. Similar findings were noted in a qualitative study of Australian emergency nurses who did not perceive manual restraint for minor injuries care as 'formal' restraint.[28]

Our findings reflect the high frequency of BtC in the ED reported in other literature.[13] While staff felt that that some incidents of BtC could be prevented, this was mainly through resource-intense approaches (ie, 1:1 care or a dedicated space for triage and assessment), which participants felt was unlikely to prove realistic in the context of resource limitations. The under-use of family and friends was noted in research prior to COVID-19,[29] and is likely to have been exacerbated during and after the pandemic.

Participants in our study expressed support for a specific checklist or flow chart approach to guide care and behaviour management in the ED. These have been used successfully in a variety of healthcare settings to act as an aid to decision-making or memory recall, including responding to BtC for people living with dementia in residential care.[30 31]

This study has important implications for practice. Findings suggest the need for regular site-wide dementia education for all staff, including information specific to the ED environment and better education on the approaches that can be used to prevent and de-escalate BtC. They also highlight the need for evidence-based resources to support this population of patients in the ED, and greater collaboration with family members and carers, to prevent unnecessary use of sedation and restraint.

## CONCLUSION

The findings of this study suggest that ED staff do not feel they are prepared to effectively respond to BtC in the ED due to the unique combination of triggers presented by the environment, and the lack of resources available in this setting. This is an increasing issue, due to the rising numbers of ED admissions for people living with dementia. Future work could adapt or develop an intervention to effectively support ED staff in managing BtC associated with dementia.

**Acknowledgements** We would like to thank the ED staff who took part in interviews for this research, as well as those who promoted this study.

**Contributors** SV, SM, EC, RC, RH, HT and JRB participated in study conception, design and delivery. LG and CL participated in study coordination, performed and coded the interviews, analysed the data and drafted the manuscript. All authors were responsible for the critical revision of the manuscript for publication and approved the final version. SV acts as guarantor.

**Funding** The study was funded by the University Hospitals Bristol and Weston NHS Foundation Trust Above and Beyond (Charitable Trustees of UH Bristol) and UH Bristol Research Capability Funding (RCF) Grants awarded through the Research Funding Committee (2019-Spr-03).

**Competing interests** None declared.

**Patient and public involvement** Patients and/or the public were involved in the design, or conduct, or reporting, or dissemination plans of this research. Refer to the Methods section for further details.

**Patient consent for publication**  Not applicable.

**Ethics approval**  This study involves human participants and ethical approval was granted by the NHS Health Research Authority (20/HRA/0587) as well as the University of the West of England (UWE Bristol) Faculty of Health and Applied Sciences Research Ethics Committee (HAS-NAM-18-027). Participants gave informed consent to participate in the study before taking part.

**Provenance and peer review**  Not commissioned; externally peer reviewed.

**Data availability statement**  Data are available on reasonable request.

**ORCID iDs**
Laura Goodwin http://orcid.org/0000-0002-9118-4620
Sarah Voss http://orcid.org/0000-0001-5044-5145

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
