## [Reviewer comments · BMJ Open]

ARTICLE DETAILS

TITLE (PROVISIONAL)	How do Emergency Department staff respond to behaviour that challenges displayed by people living with dementia? A mixed-methods study
AUTHORS	Goodwin, Laura; Liddiard, Cathy; Manning, Sera; Bengler, Jonathan; Carlton, Edward; Cheston, Richard; Hoskins, Rebecca; Taylor, Hazel; Voss, Sarah

VERSION 1 – REVIEW

REVIEWER	Wang, Nianyang University of Maryland at College Park, Health Policy and Management
REVIEW RETURNED	12-May-2023

GENERAL COMMENTS	The authors of this paper produced an mixed methods analysis regarding ED staff experiences with patients living with dementia. I applaud the authors for researching the vital topic of documenting and creating health care plans for the growing dementia population which is necessary to ensure the wellbeing of both patients and providers. My comments are as follows: Page 4 lines 46-50: The referenced approaches to help hospital staff respond to challenging behavior are from systems outside of the UK, does the UK system (and more specifically, the hospitals of this study) currently incorporate these approaches in the ED? Page 9 lines 6-9: The use of "survey respondents" and "interviewees" can be confusing within these two sentences, I suggest using "phone interviewees" to better differentiate these two samples throughout the paper. Page 10 lines 46-50: Aside from the one participant who developed their own ED-specific dementia training, are there standardized resources that can fill this training gap for hospital staff? If so, does the responsibility of providing ED-specific dementia training fall on the NHS, the hospital staff, or another entity? Overall, the authors crafted an excellent study and I look forward to the interventions that can be implemented based on their findings.
--

REVIEWER	Yous, Marie-Lee McMaster University School of Nursing
REVIEW RETURNED	15-Jun-2023

GENERAL COMMENTS	Thank you for the opportunity to review this interesting paper exploring the experiences of staff supporting people living with dementia seeking care in the ED. This is such an important area to explore and advocate for better approaches to support this growing population. Please see comments below.  1. In the abstract can you provide a definition or examples of what you mean by "behaviour that challenges"? 2. Good background! 3. In Methods under study design can you provide more context for the study sites? How many older adults or people living with dementia access these services? How many beds? What is the average length of stay? Is this different for people living with dementia experiencing behaviour that challenges? 4. In Methods under data collection for the survey how exactly was the survey developed? Did the study team refer to specific frameworks or concepts found in the existing literature? 5. The same goes for the interview. How was the interview guide developed? Was it pilot tested? 6. The PPI section at the end of methods could be moved up when first mentioning the PPI. I was wondering who was included in this group and what were the roles much earlier. 7. In the results when presenting participant characteristics it would be helpful to include a few sentences describing the sample. 8. The results were interesting and very well presented! 9. The discussion was concisely written with good comparison of existing literature. I would have preferred a great elaboration on policy and practice implications. It would be good to talk about these all in one place. For example findings revealed the need for site-wide dementia education for staff including senior and junior staff. There is a need for education on restraint use and approaches that can be used to prevent behaviours that challenges. From a systems level there is a need for greater resources to support this population and greater collaboration with family carers regardless of whether we are in a pandemic.
---

VERSION 1 – AUTHOR RESPONSE

Reviewer	Comment	Response	Location of change
1	Page 4 lines 46-50: The referenced approaches to help hospital staff respond to challenging behaviour are from systems outside of the UK, does the UK system (and more specifically, the hospitals of this study) currently incorporate these approaches in the ED?	Using the RCN 'This is me' tool is seen as best practice in EDs in the UK and is used at all three of the participating trusts in this study. This has been added as a reference to the point about 'hospital-ready information'. We have also added a sentence to note that there is no standardised use of any of these approaches across EDs in the UK.	Background – reference 20 added, and sentence added at the bottom of page 3 Methods – last sentence of 'setting' section

	Page 9 lines 6-9: The use of “survey respondents” and “interviewees” can be confusing within these two sentences, I suggest using “phone interviewees” to better differentiate these two samples throughout the paper.	This has been changed throughout, as requested	Throughout findings – pages 6-12
	Page 10 lines 46-50: Aside from the one participant who developed their own ED-specific dementia training, are there standardized resources that can fill this training gap for hospital staff? If so, does the responsibility of providing ED-specific dementia training fall on the NHS, the hospital staff, or another entity?	There isn't any specific ED focused training in the UK. Wait et al 2020 carried out a systematic review to look for interventions and could not find focused interventions rather they are attached to more general assessments like screening for delirium (https://www.britishjournalofnursing.com/content/professional/caring-for-older-people-with-dementia-in-the-emergency-department) This is noted on page 4 – lines 39-40, and is one of the main points leading to the need for this study.	N/A – detailed on page 4, lines 39-40
2	In the abstract can you provide a definition or examples of what you mean by "behaviour that challenges"?	A couple of sentences have been added to the objectives section of the abstract to address this point	Abstract - objectives
	In Methods under study design can you provide more context for the study sites? How many older adults or people living with dementia access these services? How many beds? What is the average length of stay? Is this different for people living with dementia experiencing behaviour that challenges?	Most of the information requested is not available without further research/audit. However we have provided some contextual information about the sites which will hopefully give a better idea of the study setting in relation to dementia and ED attendance.	Methods – setting section – top of page 4

	In Methods under data collection for the survey how exactly was the survey developed? Did the study team refer to specific frameworks or concepts found in the existing literature?	This detail has been added.	Methods – survey – bottom of page 4
	The same goes for the interview. How was the interview guide developed? Was it pilot tested?	More detail has been added.	Methods – interviews – top of page 5
	The PPI section at the end of methods could be moved up when first mentioning the PPI. I was wondering who was included in this group and what were the roles much earlier.	This has been moved to be the second part of the methods section	Methods – Patient and Public Involvement – top of page 4
	In the results when presenting participant characteristics it would be helpful to include a few sentences describing the sample.	Descriptive sentences on gender and profession have been added to the results section.	Results – participant characteristics – bottom of page 5
	The discussion was concisely written with good comparison of existing literature. I would have preferred a great elaboration on policy and practice implications. It would be good to talk about these all in one place. For example findings revealed the need for site-wide dementia education for staff including senior and junior staff. There is a need for education on restraint use and approaches that can be used to prevent behaviours that challenges. From a systems level there is a need for greater resources to support this population and greater collaboration with family carers regardless of whether we are in a pandemic.	Thank you. We have added a short paragraph at the end of the discussion section to summarise the implications for practice.	Discussion – bottom of page 13

VERSION 2 – REVIEW

REVIEWER	Wang, Nianyang University of Maryland at College Park, Health Policy and Management
REVIEW RETURNED	25-Jul-2023

GENERAL COMMENTS	The authors produced an outstanding revision. I have no further comments.
---

REVIEWER	Yous, Marie-Lee McMaster University School of Nursing
REVIEW RETURNED	13-Jul-2023

GENERAL COMMENTS	Thank you to the authors for taking the time to carefully incorporate feedback provided by the reviewers. I have no further suggestions.
--